# Diurnal Temperature Ranges in Relation to Lower Limb Amputation Rate of Diabetic Foot in South Korea: A Population Based Nationwide Study

**DOI:** 10.3390/ijerph18179191

**Published:** 2021-08-31

**Authors:** Sung Hun Won, Hyung-Jin Chung, Jinyoung Lee, Ye Jin Jeon, Dong-Il Chun, Tae-Hong Min, Jaeho Cho, Sungho Won, Young Yi

**Affiliations:** 1Department of Orthopedic Surgery, Soon Chun Hyang University Seoul Hospital, Seoul 04401, Korea; orthowon@schmc.ac.kr (S.H.W.); orthochun@gmail.com (D.-I.C.); minth916@gmail.com (T.-H.M.); 2Department of Orthopedic Surgery, Inje University Sanggye Paik Hospital, Seoul 01757, Korea; orthoman@paik.ac.kr; 3Department of Statistics, Chung-Ang University, Seoul 06974, Korea; joa.young424@gmail.com; 4RexSoft Corps, Seoul 08826, Korea; jeonye1028@yuhs.ac (Y.J.J.); sunghow@gmail.com (S.W.); 5Department of Public Health, Yonsei University Graduate School, Seoul 03722, Korea; 6Department of Orthopaedic Surgery, Hallym University Chuncheon Sacred Heart Hospital, Chuncheon 24253, Korea; hohotoy@nate.com; 7Graduate School of Public Health, Seoul National University, Seoul 08826, Korea; 8Department of Orthopedic Surgery, Inje University Seoul Paik Hospital, Seoul 04551, Korea

**Keywords:** diabetic foot, diabetic peripheral neuritis, diurnal temperature range, amputation

## Abstract

The evidence for the association between diurnal temperature range (DTR) and diabetic foot amputations is limited. We aimed to investigate the region-specific association between DTR and the amputation rate of diabetic foot in Korean national-wide data. Daily data on DTR and the rate of diabetic foot amputations from 16 provincial capital cities in Korea were obtained (2011–2018). In this study, the latitude ranged from 33°11′ N to 38°61′ N, and we classified each region according to latitude. Region 1, which was located at a relatively high latitude, included Seoul, Incheon, Gyeonggi-do, and Gangwon-do. Region 2, which was located at a relatively low latitude, included Busan, Ulsan, Gyeonsannam-do, Gwangju, Jeollanam-do, Jeollabuk-do, and Jeju-do. The region-specific DTR effects on the amputation rate were estimated based on a quasi-Poisson generalized linear model, combined with a distributed lag non-linear model based on the self-controlled case series design. The DTR impacts were generally limited to a period of nine days, while significant effects during lag days 7–14 were only found in the cities of Seoul, Incheon, and Gyeonggi-do (10th lag day: RR [95% CI]; Seoul: 1.015, [1.001–1.029]; Incheon: 1.052 [1.006–1.101]; Gyeonggi-do: 1.018 [1.002–1.034]). In the subgroup analysis (according to the latitude), an increase of 1 °C in DTR was associated with the risk of diabetic foot in relatively high latitude regions. DTR has considerable effects on the risk of diabetic foot amputation in various provinces in Korea, and it was particularly affected by latitude. The results can inform the decisions on developing programs to protect vulnerable subpopulations from adverse impacts.

## 1. Introduction

The burden of chronic diseases has gradually increased over the past few decades, as life expectancy increases. The number of diabetic patients is increasing exponentially as well. A typical complication of diabetic patients is diabetic foot, which occurs in at least 15% to 25% of diabetic patients in their lifetime. It has been reported that every 30 s, a lower limb is lost somewhere in the world as a consequence of diabetes, and the annual amputation rate of diabetic foot ulcers increased from 0.95% in 2012 to 1.10% in 2016 in South Korea. Diabetic foot is one of the major causes of socioeconomic loss. Therefore, it is necessary to prevent the exacerbation of diabetic foot by controlling the environmental factors that may lead to diabetic foot amputation [1,2].

Temperature has been variously analyzed as a risk factor associated with exacerbation and the amputation of diabetic foot wounds. Diabetic foot temperature may easily reach hazard thresholds, due to chronic inflammation from prolonged stress, and impairment in temperature regulation from autonomic neuropathy [3]. The ambient body surface temperature has been considered as an important risk factor in the amputation of diabetic foot [4]. A number of related studies have been conducted, and it is believed that the risk of amputation increases when the body surface heat is reduced. One study analyzes the seasonal effects of the amputation of diabetic foot. Although studies related to the amputation of diabetic foot, or delayed wound recovery, and diurnal temperature range have been conducted in animal studies [5], additional studies have not yet been reported.

Diurnal temperature range (DTR), an important indicator of climate change and variability, is an index of within-day temperature variations and has been reported to have significant adverse effects on health outcomes, including mortality due to chronic obstructive pulmonary disease [6,7], stroke [8,9], cardiovascular disease [10,11], and admissions for respiratory tract infections [12,13].

It has been reported that DTR is particularly associated with the occurrence and exacerbation of vascular disease. In addition to cerebrovascular disease and cardiovascular disease, associations with multiple sclerosis have also been reported. S. Zheng et al. [14]. analyzed 46,609 patients in northwestern China and said that DTR results in changes in blood pressure. In particular, systolic blood pressure and peripheral blood pressure were highly correlated, and the association between DTR and blood pressure varied significantly by education level. Thus, general health education might help reduce the risks of climate-sensitive diseases among high-risk groups when facing temperature changes [14].

Diabetic foot is a microvascular pathology disease. A decrease in blood flow, due to DTR, could delay wound healing and weaken the tissue’s defense mechanisms against infection [15]. These findings suggest that diurnal temperature range could be a triggering factor for the acute exacerbation of diabetic foot. However, little is known regarding the relationship between DTR and the risk of the acute aggravation of diabetic foot.

Therefore, this study aimed to investigate the association between DTR and the risk of the acute exacerbation of diabetic foot requiring major or minor amputation in South Korea, using a time-stratified case crossover design. We anticipate that the larger the DTR, the longer the wound healing period will be, thus aggravating the diabetic foot wounds. We assume that this study will be a basis for the establishment of policy for managing diabetic foot patients by identifying the relationship between DTR and the amputation rate in diabetic foot patients.

## 2. Materials and Methods

### 2.1. Study Population

In this nationwide population-based retrospective study, we investigated the data of 420,096 DM patients aged ≥18 years, using the Korean Health Insurance Review and Assessment Service (HIRA) claims database. The medical codes that signify diabetic amputation were implemented in January 2011. Therefore, the current study included data from 2011 to 2018. During the research period, we included patients in the analysis after they had gone through a wash-out period of one year. After the HIRA claims database merged with DTR data, 8156 patients who were assigned the amputation practice code after the diagnosis of DM were included in the main analysis. The definition of a DM diagnosis, based on the International Classification of Diseases 10th revision (ICD-10) codes (E10, E11, E12, E13, E14) is the principal diagnosis or additional diagnosis of diabetes, in addition to being prescribed at least one antidiabetic drug (including glucose-lowering drugs and insulin) in a given year. An antidiabetic drug was defined using Anatomical Therapeutic Chemical (ATC) codes (glucose-lowering drug: A10B; insulin: A10A) and a medical review. The Institutional Review Board of Seoul Paik Hospital approved the present retrospective cohort study (IRB approval no., PAIK 2020-04-013).

### 2.2. Diurnal Temperature Range (DTR) of Each Region

The definition of DTR (daily) is the difference between the maximum and minimum daily temperature range. To define DTR, an automated surface observed the temperature information in 16 provincial capital cities, using the weather data service of the Korea Meteorological Administration (KMA) (data.kma.go.kr). Local information on the medical care/treatment institution from the HIRA claims data was combined with the observed daily temperature to analyze the risk of diabetic foot amputation in each region. DTR (daily) is included in the analysis as a continuous variable, according to previous findings and sensitivity analyses [16].

### 2.3. Definition of Diabetic Foot Amputation/Ulcer

The main outcome (diabetic foot amputation related with PAD) was not defined as a disease code. Therefore, diabetic foot amputation was defined manipulatively based on the medical practice codes, including percutaneous transluminal angioplasty in another location (M6597), the percutaneous intravascular installation of a metallic stent in another location (M6605), percutaneous intravascular atherectomy (M6620), and the amputation of extremities (N0571–N0575). In addition, amputation after the index date of DM was considered as the occurrence of an event, and the earliest amputation was classified as the index event.

### 2.4. Study Area

This study included 16 provincial capital cities in South Korea between the longitude from 124°60′ E to 131°87′ E and the latitude from 33°11′ N to 38°61′ N. For subgroup analyses, we classified each region according to latitude. Region 1, which was located at a relatively high latitude, included Seoul, Incheon, Gyeonggi-do, and Gangwon-do. Region 2, which was located at a relatively low latitude, included Busan, Ulsan, Gyeonsannam-do, Gwangju, Jeollanam-do, Jeollabuk-do, and Jeju-do.

### 2.5. Statistical Analyses

All analyses were conducted in R (version 3.6.3) (Rexsoft, Co. Ltd., Seoul, Korea). In this study, we selected self-controlled case series (SCCS) with a lag structure, to estimate the delayed effect (from 1 to 14 days) of DTR (daily) on diabetic foot amputation, and these delayed times are defined as lag days. The association between DTR (per 1 °C increase) and the risk of diabetic foot amputation (between 1 and 14 lag days) was estimated with the SCCS model. By using the SCCS model, as it is a self-controlled method, each patient was compared with themselves as controls to minimize the impact of time-invariant confounders [17]. As DTR is quantitative data regarding individual and environmental exposure, in this case, we used a Poisson generalized nonlinear model to analyse the SCCS model [18]. We conducted subgroup analyses, according to the relative latitude with the same statistical models. The “Kormaps” and “tmap” packages [19] were used to map the risk of diabetic foot amputation per region (16 cities) and sub-region (region 1 and 2).

## 3. Results

### 3.1. Association between DTR and Amputation of Diabetic Foot in Each Region

Table 1 and Figure 1 show the association between increased DTR and the risk of diabetic foot amputation across 14 lag days (range of lag days: 1–14) in each region. Overall, the estimated risk ratio of DTR and diabetic foot amputation was not linear by lag days, and no constant direction was observed. However, in relatively high latitude regions, including Seoul, Incheon, Gyeonggi-do, and Gangwon-do, the increasing DTR was associated with a higher risk of diabetic foot amputation, and was statistically significant in some lag days (10th lag day: RR [95% CI]; Seoul: 1.015, [1.001–1.029]; Incheon: 1.052 [1.006–1.101]; Gyeonggi-do: 1.018 [1.002–1.034]; Gangwon-do: 1.002 [0.958–1.048]). On the other hand, in relatively low latitude regions, including Busan, Ulsan, Gwangju, and Jeolla-do, the association between increasing DTR and diabetic foot amputation was not significant, or the opposite trend was observed (10th lag day: RR [95% CI]; Busan: 1.005 [0.978–1.033]; Ulsan: 1.009 [0.961–1.060]; Gwangju: 0.997 [0.966–1.029]; Jeollanam-do: 0.965 [0.918–1.016]; Jeollabuk-do: 0.995 [0.964–1.027]). This latitude-related tendency of the amputation risk supports our decision to conduct subgroup analyses in the high and low latitude regions.

### 3.2. Association between DTR and Amputation of Diabetic Foot in Sub-Region According to Latitude

The results from the subgroup analyses in region 1 (Seoul, Incheon, Gyeonggi-do, and Gangwon-do) are shown in Table 2 and Figure 2. Increasing DTR was associated with a higher risk of amputation in most lag days. On the 10th and 14th lag day, the association was statistically significant (10th lag day: 1.017 [1.007–1.027]; 14th lag day: 1.012 [1.002–1.022]). The results in region 2 (Busan, Ulsan, Gyeonsannam-do, Gwangju, Jeollanam-do, Jeollabuk-do, and Jeju-do) are shown in Table 3 and Figure 3. Differing from the results in region 1, there was no significant association between DTR and the amputation of diabetic foot, with the exception of the 3rd lag day (3rd lag day: 0.976 [0.961–0.992]).

## 4. Discussion

To the best of our knowledge, this is the first study to analyze the association between DTR and diabetic foot, including infection, ulceration, or the destruction of tissues.

The relationship between diabetic foot and plantar temperature has been analyzed in various ways in other studies. Increased plantar temperatures in individuals with a history of ulcers may be a result of acute temperature increases from plantar stresses, chronic inflammation from prolonged stresses, and impairment in temperature regulation from autonomic neuropathy [3]. Therefore, plantar temperature acts as a tool to aid the early diagnosis and exacerbation of diabetic foot. However, very few studies have analyzed whether changes in the temperature or environmental factors can aggravate diabetic foot.

Based on the national scale of this study, the estimated risk ratio of diurnal temperature and the amputation of diabetic foot was not linear by lag days, and no constant direction was observed. Although South Korea has a small national territory, it has the characteristic of showing a large DTR by region, due to geographical factors. Therefore, there may be no significant correlation on the national scale.

The correlation between DTR and diabetic foot amputation was analyzed in both region 1 and region 2. Region 1 (Seoul, Incheon, Gangwon-do, and Gyeonggi-do) is less affected by the sea and is located at a relatively high latitude, while region 2 (Busan–Ulsan–Gwangju–Jolla) is adjacent to the sea and is located at a relatively low latitude. The risk of amputation, related to the change in DTR, showed a significant correlation between lag days 10 and 14 in region 1. However, there was no significant correlation in region 2. In South Korea, low-latitude regions have a close relationship with the coast, meaning the DTR difference was relatively small compared to the high-latitude regions. Based on this result, it can be inferred that the exacerbation of diabetic foot under geopolitics is affected by DTR. The delay period of approximately 10 to 14 days is thought to be caused by the wound recovery process not being performed properly, and the time it takes for clinical symptoms to appear.

Won et al. [5] showed that the wound healing period was significantly longer in the group with the large DTR. The thickness of the scar, and the number of angiogenesis, were relatively small, although there was no statistical significance. It is predicted that the change in DTR would play a major role in the inflammatory response period of wound healing. We predict that the larger the DTR, the slower the wound healing will occur.

HB leung et al. [20] analyzed seasonal variations in the non-traumatic major lower limb amputation of diabetic patients living in Hong Kong. It was confirmed that the need for major amputation was repeatedly impacted by seasonal factors, particularly in early spring, followed by amputation in summer. Wound aggravation in the summer is explained by the high temperatures that may worsen wound hygiene, which leads to infection. There was no other explanation in early spring, and it was reported that diabetic foot ulcer management should be thoroughly managed, and an extensive program should be delivered in early spring. Jehan et al. [21] also reported seasonal variations in hospitalization due to diabetic foot complications. Intensive patient education and the clustering of medical services during the winter months may help to lower the prevalence of diabetic foot ulcers and, as a result, amputations. They claimed that various complications arose over the winter, as well as a lengthy stay in the hospital. HB leung’s study, which directly measured the number of amputations, reported that there were more amputations in early spring. These characteristics suggest that changes in DTR, as well as low temperatures, may result in major diabetic foot amputation.

This study has several limitations. First, since HIRA provides only the location of the medical institutions where the amputation was performed, DTR was also calculated based on the location of the medical institution, not the individual’s residence, meaning the correlation may be low if patients traveled a long distance for surgery. Therefore, the actual DTR acting on the patient may be different from the measurement, because the measuring station is administratively located in each region. Second, there is a possibility that the amputation surgery was delayed due to vascular intervention. Third, in the current study, we could not include seasonal factor in the analysis. Because the distribution of DTR could be related with seasonality, we plan to consider seasonality in future investigations. Fourth, in this study, we used a self-controlled model with a SCCS design to control for individual level confounders, such as educational level, underlying medication, and diseases history. However, the individual level confounder could not be directly adjusted in the model, meaning there could be a remaining confounder bias. Further investigation is needed to control more environmental information and individual-level variables. Lastly, South Korea’s land area is approximately 100,210 km^2^. The latitude difference between regions is relatively small, and various climates did not appear in the study.

In this study, we have not found statistically significant or robust results of DTR on diabetic foot amputation or ulcers. Nevertheless, this study is meaningful in that it is the first study in Korea to explore the link between DTR and diabetic foot amputation. Based on the current results as pilot study, we will design a future study that could serve as a basis for establishing policies for managing diabetic foot patients, by identifying the relationship between DTR and the amputation rate in diabetic foot patients.

## 5. Conclusions

Increases in DTR are associated with a higher risk of amputation, particularly in relatively high latitude regions in South Korea. This is the first study in South Korea to estimate the association between DTR and diabetic foot amputation, while considering relative latitude. This study is clinically relevant in the way that it can be applied in daily living, by informing diabetic foot patients and clinicians living in high latitude areas that DTR is also a variable to consider for wound exacerbation and amputation potential. In addition, an education center can be built in these high risk regions for preventing the exacerbation of diabetic foot. Further investigation is needed to find robust findings and to discuss the biological mechanism. The study may become a cornerstone for future developments in reducing the amputation rate of diabetic foot patients, through the control of living environments using IoT technology.

## Figures and Tables

**Figure 1 ijerph-18-09191-f001:**
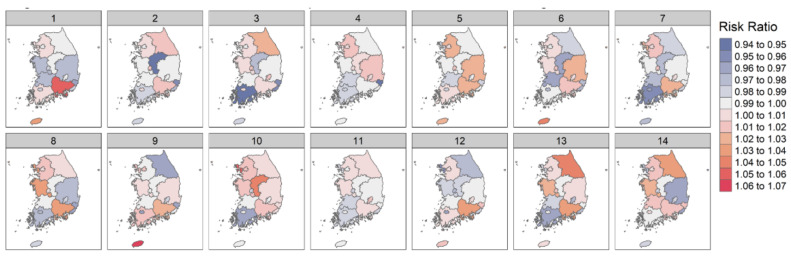
Association between DTR and amputation of diabetic foot in each region.

**Figure 2 ijerph-18-09191-f002:**
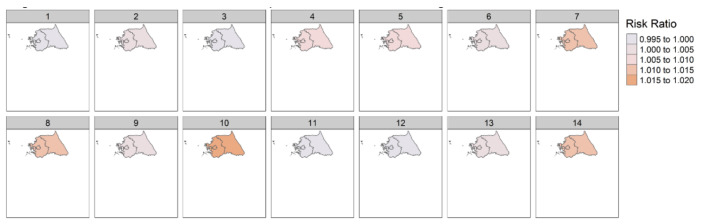
Association between DTR and amputation of diabetic foot in region 1.

**Figure 3 ijerph-18-09191-f003:**
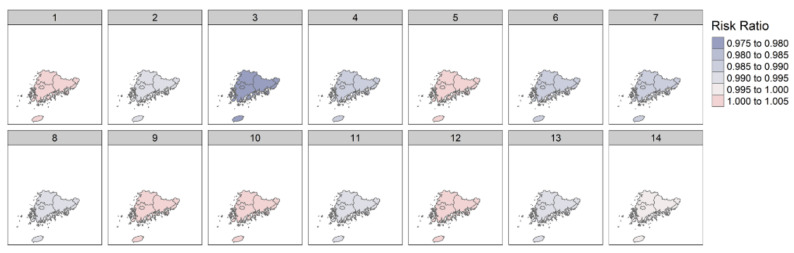
Association between DTR and amputation of diabetic foot in region 2.

**Table 1 ijerph-18-09191-t001:** Association between DTR and risk of diabetic foot amputation in each region.

Region	Risk Ratio (95% Confidence Interval) along Lag Day (Range: 1–14) ^(1)^													
	1	2	3	4	5	6	7	8	9	10	11	12	13	14
Seoul	0.997 (0.983–1.011)	1.000 (0.986–1.014)	0.993 (0.979–1.007)	1.001 (0.987–1.015)	1.002 (0.988–1.016)	1.000 (0.986–1.014)	1.007 (0.993–1.021)	1.008 (0.994–1.022)	1.013 (0.999–1.027)	1.015 (1.001–1.029) *	0.996 (0.982–1.01)	1.001 (0.987–1.015)	1.005 (0.991–1.019)	1.010 (0.995–1.023)
Incheon	1.003 (0.959–1.049)	0.999 (0.955–1.045)	1.000 (0.956–1.046)	1.003 (0.959–1.049)	0.988 (0.945–1.034)	1.010 (0.966–1.057)	1.024 (0.978–1.07)	1.034 (0.989–1.082)	1.017 (0.972–1.064)	1.052 (1.006–1.101) *	0.982 (0.939–1.027)	0.959 (0.917–1.003)	0.990 (0.946–1.036)	0.977 (0.933–1.021)
Gyeong-gi	1.006 (0.99–1.022)	1.007 (0.991–1.023)	1.007 (0.991–1.023)	1.017 (1.001–1.033) *	1.021 (1.004–1.036) **	1.010 (0.994–1.026)	1.017 (1.001–1.033) *	1.012 (0.996–1.028)	0.999 (0.983–1.015)	1.018 (1.002–1.034) *	1.004 (0.988–1.02)	1.003 (0.987–1.019)	1.003 (0.987–1.019)	1.017 (1.001–1.033) *
Busan	0.988 (0.961–1.016)	0.980 (0.954–1.007)	0.954 (0.928–0.981) ***	0.994 (0.967–1.022)	0.999 (0.972–1.027)	0.979 (0.953–1.006)	0.985 (0.96–1.011)	0.970 (0.943–0.996) *	0.994 (0.967–1.022)	1.005 (0.978–1.033)	0.990 (0.963–1.018)	0.985 (0.958–1.013)	0.971 (0.944–0.997) *	0.969 (0.944–0.994) *
Daegu	0.998 (0.975–1.022)	0.998 (0.975–1.022)	1.013 (0.99–1.037)	1.006 (0.983–1.03)	0.998 (0.975–1.022)	1.005 (0.982–1.029)	1.005 (0.982–1.029)	0.988 (0.965–1.012)	0.982 (0.959–1.006)	0.998 (0.975–1.022)	1.012 (0.989–1.036)	1.007 (0.984–1.031)	0.978 (0.955–1.001)	0.987 (0.964–1.011)
Daejeon	0.986 (0.955–1.016)	1.016 (0.985–1.048)	1.016 (0.985–1.048)	1.007 (0.976–1.039)	1.013 (0.982–1.045)	1.018 (0.987–1.051)	1.020 (0.988–1.052)	1.007 (0.976–1.039)	0.998 (0.967–1.03)	0.989 (0.959–1.021)	0.981 (0.95–1.011)	0.979 (0.949–1.01)	1.015 (0.983–1.046)	0.986 (0.956–1.018)
Gwangju	0.986 (0.954–1.02)	1.005 (0.974–1.037)	1.001 (0.97–1.033)	0.985 (0.954–1.015)	0.990 (0.959–1.022)	0.973 (0.941–1.006)	0.962 (0.93–0.994) *	0.98 (0.948–1.013)	0.962 (0.93–0.994) *	0.997 (0.966–1.029)	0.993 (0.962–1.025)	0.965 (0.934–0.998) *	0.998 (0.967–1.03)	1.009 (0.978–1.041)
Ulsan	0.967 (0.919–1.017)	0.960 (0.913–1.011)	0.953 (0.905–1.002)	0.948 (0.900–0.997) *	0.983 (0.936–1.033)	0.978 (0.931–1.026)	0.996 (0.948–1.046)	0.967 (0.919–1.017)	0.962 (0.914–1.012)	1.009 (0.961–1.06)	0.994 (0.945–1.046)	1.016 (0.968–1.067)	0.996 (0.948–1.046)	1.018 (0.969–1.069)
Kyeonsannam-do	1.056 (1.026–1.088) ***	1.017 (0.988–1.047)	1.007 (0.978–1.037)	1.000 (0.969–1.032)	1.024 (0.992–1.056)	1.014 (0.982–1.045)	1.025 (0.996–1.056)	1.035 (1.005–1.065) *	1.028 (0.998–1.058)	1.015 (0.986–1.045)	1.005 (0.974–1.037)	1.035 (1.003–1.068) *	1.035 (1.004–1.069) *	1.020 (0.991–1.051)
Kyeonsanbuk-do	0.976 (0.937–1.017)	0.997 (0.957–1.039)	0.993 (0.953–1.035)	1.019 (0.978–1.062)	1.026 (0.985–1.069)	1.03 (0.989–1.074)	0.999 (0.959–1.041)	0.976 (0.937–1.017)	1.002 (0.962–1.044)	1.01 (0.969–1.052)	0.992 (0.952–1.034)	0.992 (0.952–1.034)	1.008 (0.967–1.05)	0.967 (0.928–1.007)
Jeollanam-do	1.008 (0.958–1.061)	0.989 (0.94–1.041)	0.945 (0.896–0.996)	0.984 (0.935–1.036)	1.006 (0.956–1.059)	1.000 (0.95–1.052)	0.958 (0.91–1.008)	0.98 (0.931–1.031)	1.013 (0.965–1.064)	0.965 (0.918–1.016)	0.990 (0.941–1.042)	1.003 (0.953–1.055)	0.967 (0.919–1.017)	0.991 (0.942–1.043)
Jeollabuk-do	0.971 (0.939–1.003)	0.994 (0.963–1.026)	0.993 (0.962–1.025)	0.985 (0.954–1.015)	0.981 (0.949–1.014)	0.962 (0.930–0.994) *	0.971 (0.939–1.003)	0.999 (0.966–1.033)	0.994 (0.963–1.026)	0.995 (0.964–1.027)	0.987 (0.957–1.019)	0.994 (0.963–1.026)	0.988 (0.956–1.022)	1.012 (0.981–1.044)
Chungcheongnam-do	0.978 (0.935–1.023)	0.995 (0.951–1.041)	1.001 (0.957–1.047)	0.994 (0.95–1.04)	0.996 (0.952–1.042)	1.004 (0.96–1.05)	0.993 (0.949–1.039)	1.034 (0.99–1.079)	1.006 (0.964–1.05)	1.014 (0.971–1.059)	1.005 (0.961–1.051)	0.99 (0.946–1.036)	1.023 (0.978–1.07)	1.021 (0.978–1.066)
Chungcheongbuk-do	0.991 (0.953–1.031)	0.945 (0.907–0.984) **	0.974 (0.936–1.012)	1.004 (0.965–1.044)	0.999 (0.959–1.041)	0.962 (0.923–1.002)	0.977 (0.94–1.016)	0.995 (0.957–1.035)	0.992 (0.954–1.032)	1.042 (1.002–1.084) *	0.993 (0.954–1.032)	0.989 (0.949–1.031)	0.992 (0.952–1.034)	0.991 (0.953–1.031)
Gangwon-do	0.991 (0.947–1.037)	1.014 (0.969–1.061)	1.024 (0.979–1.072)	1.000 (0.956–1.046)	0.996 (0.952–1.042)	0.987 (0.944–1.033)	0.987 (0.944–1.033)	1.005 (0.961–1.051)	0.965 (0.923–1.01)	1.002 (0.958–1.048)	1.008 (0.964–1.055)	0.979 (0.935–1.023)	1.042 (0.996–1.09)	1.039 (0.995–1.085)
Jeju-do	1.037 (0.958–1.125)	0.987 (0.907–1.074)	0.982 (0.903–1.069)	0.997 (0.917–1.081)	1.021 (0.942–1.107)	1.044 (0.961–1.134)	0.975 (0.895–1.063)	0.986 (0.905–1.075)	1.062 (0.982–1.148)	0.998 (0.919–1.084)	0.996 (0.917–1.081)	1.011 (0.933–1.096)	1.001 (0.92–1.089)	0.984 (0.903–1.073)

^(1)^ Risk ratio, estimated from a quasi-Poisson generalized linear model, combined with a distributed lag non-linear model, based on the self-controlled case series design. *: *p* value < 0.10; **: *p* value < 0.05; ***: *p* value < 0.001.

**Table 2 ijerph-18-09191-t002:** Association between DTR and risk of diabetic foot amputation in region 1.

	Risk Ratio (95% Confidence Interval) Along Lag Day (Range: 1–14) ^(1)^													
	1	2	3	4	5	6	7	8	9	10	11	12	13	14
Region 1 ^(2)^	1.000 (0.990–1.010)	1.003 (0.993–1.013)	1.000 (0.990–1.010)	1.008 (0.998–1.018)	1.009 (0.999–1.019)	1.004 (0.994–1.014)	1.011 (1.001–1.021)	1.011 (1.001–1.021)	1.005 (0.995–1.015)	1.017 (1.007–1.027)	0.999 (0.989–1.009)	0.999 (0.989–1.009)	1.005 (0.995–1.015)	1.012 (1.002–1.022)

^(1)^ Risk ratio, estimated from a quasi-Poisson generalized linear model, combined with a distributed lag non-linear model, based on the self-controlled case series design. ^(2)^ Region 1 included Seoul, Incheon, Gyeonggi, and Gangwon province.

**Table 3 ijerph-18-09191-t003:** Association between DTR and risk of diabetic foot amputation in region 2.

	Risk Ratio (95% Confidence Interval) Along Lag Day (Range: 1–14) ^(1)^													
	1	2	3	4	5	6	7	8	9	10	11	12	13	14
Region 2 ^(2)^	1.003 (0.987–1.019)	0.991 (0.976–1.007)	0.976 (0.961–0.992)	0.988 (0.973–1.004)	1.001 (0.985–1.017)	0.988 (0.973–1.004)	0.99 (0.975–1.006)	0.993 (0.978–1.009)	1.003 (0.987–1.019)	1.002 (0.986–1.018)	0.994 (0.979–1.01)	1.005 (0.989–1.021)	0.993 (0.978–1.009)	0.997 (0.981–1.013)

^(1)^ Risk ratio, estimated from a quasi-Poisson generalized linear model, combined with a distributed lag non-linear model, based on the self-controlled case series design. ^(2)^ Region 2 included Busan, Ulsan, Kyeonsannam-do, Gwangju, Jeollanam-do, Jeollabuk-do, and Jeju-do.

## Data Availability

The data are distributed to registered users through the official website of HIRA Healthcare Bigdata Hub (https://opendata.hira.or.kr/home.do, accessed on 24 August 2020). After the evaluation of a research proposal by the HIRA review committee, registered users can receive special access privileges to the data.

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
