# Peer review of "Diurnal Temperature Ranges in Relation to Lower Limb Amputation Rate of Diabetic Foot in South Korea: A Population Based Nationwide Study"

_ijerph, 2021, doi:10.3390/ijerph18179191_

Round 1
Reviewer 1 Report
Title: remove “in” …. Lower Limb Amputation rate of “in” Diabetic Foot
Abstract:
The high and low latitudes should be introduced in the method part of the abstract.
Introduction:
It is better to show the incidence of the DM foot and the percentage of amputation in the country or regions. This may be a key to support the rationale of the study.
Row 54 – please explain the word “adverse effects” in terms of the relationship between DTR and health outcomes.
Row 64 – it is not clear that why the DTR and blood pressure varied with “the education level”?
I suggest writing more review literature between the temperature and DM foot pathology.
Please propose the hypothesis of the study.
Method:
Please provide the definition of “lag day” in the method.
Row 77 – the data were collected between 2011-2018, why 2011-2018? Any season of collecting these data and may influence the results?
Row 109 – the study area was defined as high latitude and low attitude. However, in the result part, it was presented in the median latitude (row 129&139). The definition of high latitude and median latitude was defined as the same terminology?
Results:
Row 129 and 139 – please double-check the word median latitude.
Discussion:
What is the strength of this study?
Paragraph 2 – mentioned previous studies, a reference needs to be added.
Paragraph 4 – subgroup analysis was performed in this study, and the results showed significant difference in high latitude area, but not low attitude area. However, there was no discussion or review in terms of the geography and the pathology of the DM foot. I suggest discussing in this detail.
Paragraph 6 – the temperature is one of many factors that aggravate the exacerbation of diabetic foot. This study did not control other factors. I think the authors should discuss other factors which may affect the results of this study such as living area, educational level, medication, other underlying diseases, etc.
Row 211-213 – the detail is controversial in this sentence “Intensive patient education and clustering of medical services in winter might reduce the incidence of diabetic foot ulceration, hence amputations. They reported that many complications occurred in winter and a long hospital bed period.”
Conclusion
The conclusion is quite general and did not present the summary results of the study. Please revise the conclusion and add summarized results at beginning of the conclusion.
Table
Table 2 and 3 – there is an explanation “(Note) *: p value < 0.10; **: p value < 0.05; ***: p value < 0.001172”. However, I did not see the sign in the table. The signs explanation should be removed.
Author Response
Author responses to the reviewers’ comments:
Diurnal temperature ranges in relation to Lower Limb Amputation rate of in Diabetic Foot in South Korea: a population based nationwide study
We sincerely thank the editor and reviewers for their insightful and constructive comments. We have considered all comments and responded to them below. We hope that the manuscript is now deemed suitable for publication in The International Journal of Environmental Research and Public Health.
Thank you again for the careful and helpful review.
Reviewers’ Comments:
Title: remove “in” …. Lower Limb Amputation rate of “in” Diabetic Foot
- Thank you for the comment. The title has been modified as follows. “Diurnal temperature ranges in relation to Lower Limb Amputation rate of Diabetic Foot in South Korea: a population based nationwide study”
Abstract:
The high and low latitudes should be introduced in the method part of the abstract.
- Thank you for the comment. We have added following paragraph to the abstract.
(Abstract, page 1)
In this study, latitude ranges from 33·11N to 38·61N, and we classified each region according to latitude. Region 1, which located at relatively high latitude, included Seoul, Incheon, Gyeonggi-do, and Gangwon-do. Region 2, which located at rela-tively low latitude, included Busan, Ulsan, Gyeonsannam-do, Gwangju, Jeollanam-do, Jeollabuk-do, and Jeju-do.
Introduction:
It is better to show the incidence of the DM foot and the percentage of amputation in the country or regions. This may be a key to support the rationale of the study.
- Thank you for the comment. We added incidence of diabetic foot amputation in Korea as below to emphasize the seriousness of the disease.
(Introduction, page 1-2, line 44-47)
It has been reported that every 30 seconds a lower limb is lost somewhere in the world as a consequence of diabetes, and the annual amputation rate of diabetic foot ulcer was in-creased from 0.95% in 2012 to 1.10% in 2016 in South Korea.
Row 54 – please explain the word “adverse effects” in terms of the relationship between DTR and health outcomes.
- Thank you for the comment. Diurnal temperature range (DTR) has been suggested to be an adverse health factor related to COPD, respiratory tract infection, stroke, and cardiovascular disease. The goal of our study was to explore the adverse effect of DTR in diabetic foot amputation rate in South Korea.
Row 64 – it is not clear that why the DTR and blood pressure varied with “the education level”?
I suggest writing more review literature between the temperature and DM foot pathology.
Please propose the hypothesis of the study.
- Thank you for the comment. Zheng et al.[14] revealed in his research article that people with lower education level were more susceptible to temperature change to increase BP. Thus, general health education might help reduce the potential risk of climate-sensitive diseases among high-risk groups when facing temperature changes.
- In response to the comment, we added information on how the increase in temperature affects diabetic foot.
(Introduction, page 2, line 51-53)
Diabetic foot temperature may easily reach hazard thresholds due to chronic inflammation from prolonged stress, and impairment in temperature regulation from autonomic neuropathy
- We have added hypothesis in the last paragraph of Introduction section.
(Introduction, page 2, lines 81-85)
We anticipate larger the DTR, wound healing period will be much longer thus aggravat-ing the diabetic foot wounds. We assume this study will be basis for policy establishment of managing diabetic foot patients by identifying the relationship between DTR and the amputation rate in diabetic foot patients
Materials and Methods/Results
- Please provide the definition of “lag day” in the method.
- Thank you for the comment. We added the description of “lag day” in the Materials and Methods section as below.
(Materials and Methods, page 3, lines 116-119)
In this study, we selected self-controlled case series (SCCS) with the lag structure to estimate the delayed from 1 to 14 days’ effect of DTR (daily) on diabetic foot amputation, and these delayed times are defined as lag days.
- Row 77 – the data were collected between 2011-2018, why 2011-2018? Any season of collecting these data and may influence the results?
- Thank you for the comment. There are two reasons why the research period is from 2011 to 2018. First, in South Korea, medical codes that signify diabetic foot or amputation were applied in January, 2011 according to implementation of the sixth edition of Korean statistical classification of disease and related health problems-6 system (KCD-6). Second, when we applying for the claim data in Korean Health Insurance Review and Assessment Service (HIRA) claim database, only data up to 2018 could be available. We further described about study period in then Materials and Methods section as below.
(Materials and Methods, page 2, lines 77-79)
Due to medical codes that signify diabetic amputation were applied in January 2011, claimed data between 2011 and 2018 included in the current study.
- Row 109 – the study area was defined as high latitude and low attitude. However, in the result part, it was presented in the median latitude (row 129&139). The definition of high latitude and median latitude was defined as the same terminology?
4. Row 129 and 139 – please double-check the word median latitude.
- Thank you for the comments. We modified incorrect wording in Methods and Results section as below.
(Materials and Methods, page 3, line 133.)
However, in relatively high latitude regions-.
(Results, page 4, lines 142-143.)
This latitude-related tendency of the amputation risk supported our decision to conduct subgroup analysis in the high and low latitude regions.
Discussion:
What is the strength of this study?
Paragraph 2 – mentioned previous studies, a reference needs to be added.
- Thank you for the comment. A reference has been added to the mentioned paragraph.
Paragraph 4 – subgroup analysis was performed in this study, and the results showed significant difference in high latitude area, but not low attitude area. However, there was no discussion or review in terms of the geography and the pathology of the DM foot. I suggest discussing in this detail.
- Thank you for the comment. In South Korea, low-latitude regions have a close relationship with the coast, so the DTR difference was relatively small compared to high-latitude regions. Therefore, correlation between the diabetic foot wound exacerbation and the DTR was insignificant in low low latitude region 2(Busan, Ulsan, Gyeonsannam-do, Gwangju, Jeollanam-do Jeollabuk-do, and Jeju-do) The correlation between the exacerbation of diabetic foot wounds and DTR is explained in discussion, line 205-216.
Paragraph 6 – the temperature is one of many factors that aggravate the exacerbation of diabetic foot. This study did not control other factors. I think the authors should discuss other factors which may affect the results of this study such as living area, educational level, medication, other underlying diseases, etc.
- Thank you for the comment. We further discussed about confounding bias due to study design in the Discussion section as below.
(Discussion, page 10, lines 227-232.)
Third, in this study, we used a self-controlled model with SCCS design to control individual level confounders such as educational level, underlying medication and diseases history. However, the individual-level confounder could not directly adjust in the model, there could be remaining confounding bias. Further investigation is needed to control more environmental information and individual-level variables.
Row 211-213 – the detail is controversial in this sentence “Intensive patient education and clustering of medical services in winter might reduce the incidence of diabetic foot ulceration, hence amputations.
They reported that many complications occurred in winter and a long hospital bed period.”
- Thank you for the comment. It seems that the meaning may have been conveyed incorrectly due to author’s miswording. Because diabetic foot exacerbation and amputation rate increase in winter, it was said to lower the incidence by education in this season. With your advice, the sentence has been clarified as follows.
“Intensive patient education and the clustering of medical services during the winter months may help to lower the prevalence of diabetic foot ulcers and, as a result, amputations. They claimed that various complications arose over the winter, as well as a lengthy stay in the hospital.”
Conclusion
The conclusion is quite general and did not present the summary results of the study. Please revise the conclusion and add summarized results at beginning of the conclusion.
- Thank you for the comment. Summary of the result has been added to the beginning of conclusion, and content has been revised as follows.
(Conclusion, page 8, line 248-368)
Increased in DTR is associated with a higher risk of amputation, especially in rela-tively high latitude regions in South Korea. This is the first study in South Korea to esti-mate the association between DTR and diabetic foot amputation with considering relative latitude. This study is clinically relevant in a way that it can be applied in daily living, by informing diabetic foot patients and clinicians living in high latitudes that DTR is also a variable to consider for wound exacerbation and amputation potential. In addition, an education center can be built in these high risk regions for preventing exacerbation of dia-betic foot. Further investigation is needed to find robust findings and to discuss the bio-logical mechanism. The study may become a cornerstone for future development in re-ducing amputation rate of diabetic foot patients through control of living environments by combining IoT technology.
Table 2 and 3 – there is an explanation “(Note) *: p value < 0.10; **: p value < 0.05; ***: p value < 0.001172”. However, I did not see the sign in the table. The signs explanation should be removed.
- Thank you for the comment. We were planning to mark the p value at first, however, we decided to delete it.

Reviewer 2 Report
As a clinician, I am not sure what this study benefits me or my patients. The findings, as stated by the authors, have no known biological mechanism. I do not think I can change the weather.
For the minimal and inconstant differences over the different areas, I do not think I would recommend changing the weather even if I could.
My recommendation of major revision is not because I think a major revision will improve anything but the presentation but because I cannot recommend acceptance, and the article does not fit in to the Reject criteria (article has serious flaws, additional experiments needed, research not conducted correctly).
Author Response
I am grateful for your guidance and feedback.
I would like to answer the comments by concisely explaining the clinical significance of this paper.
Through this study, it is thought that difference in diurnal temperature range(DTR) affects lower extremity amputation rate in diabetic foot patients, especially in patients living in high latitude regions of Korea. Since DTR is a predictable factor, a contact network can be established to notify the risk to the diabetic foot patients to prevent lower extremity amputation. In addition, government may build patient education center for preventing diabetic foot ulceration chiefly in high risk regions.
As an extension of this study, various climatic factors and the prognostic factors of diabetic foot patients are being studied. The studies are expected to have clinical significance as they can be applied directly in daily living, such as management of room air and room temperature in patient’s residence.
This research is conducted through Korean national research fund, and it will become a cornerstone for future development in reducing amputation rate of diabetic foot patients through control of living environments by combining IoT technology.

Round 2
Reviewer 1 Report
The authors have addressed my comments and concerns appropriately. The writing and overall quality has been significantly improved.
Author Response
The authors have addressed my comments and concerns appropriately. The writing and overall quality has been significantly improved.
->Thank you again for the careful and helpful review.
Reviewer 2 Report
In this study the authors try to relate diabetic foot amputations to diurnal temperature range (DTR).
I have serious concerns with this study.
To apply an intervention we need a mechanism (See Mechanism matters, Nature Medicine volume 16, page347, 2010). The authors' suggestions and their review of the literature do little for this (Hong Kong data accepted). Furthermore, the authors, in their reply state: "The studies are expected to have clinical significance as they can be applied directly in daily living, such as management of room air and room temperature in patient’s residence".
Such a statement does not mean much. Do we need to heat the patients' environment or cool it? How was the overall relationship between maximum and minimum daily temperatures? I assume they looked at these before looking at DTR. And I assume they did not share this with us as they found no correlation
I could conceptualize low temperatures could cause increased ischaemia, just as the authors conceptualize high temperatures could related to hygiene problems (not that I’m sure why they should increase ischaemia related amputations). But as it is DTR, it is necessary to propose a mechanism why maximum/minimum temperatures to not effect amputation rate but DTR does, e.g. they feel it is hot/cold and behave accordingly even though the temperature has changed to the other extreme. I don’t think an explanation like this is serious.
The main outcome, diabetic foot amputation related with peripheral artery disease defined only with ischaemia related interventions. But according to the methods, diabetes was not a criterion. So basically this issue is relating amputations for ischaemia and not for diabetes.
Looking at the results, the effect sizes for most northern areas were around 2%, with the exception of Incheon and Gangwon-do, about 5%. By the way why not refer to northern & southern areas rather than latitude? And then the southern areas have an opposite or no effect. So would the authors recommend an intervention only in the northern regions?
It is not as if the weather is so different between northern and southern regions in South Korea (~250 km) to explain this difference or permit this type of data manipulation. Furthermore I might suggest that the northern regions are slightly different socioeconomically, a possible confounder not looked at.
If I were an officer in Seoul with funds for preventing amputations related to PVD or to diabetes, I would probably want to invest in one of the interventions with proven efficacy. A 2% increase in some regions does not provide reason for a nationwide intervention.
All in all, this finding is at most a curiosity, not supported by a mechanism, for a diagnosis not correctly defined and based on doubtful statistics at the best. The authors chose not to share the raw data with us. That is their right, but not customary nowadays, when a journal supports/recommends.
The English language usage still leaves a lot to de improved.
I can therefore not recommend publication.
Author Response
Author responses to the reviewers’ comments:
Diurnal temperature ranges in relation to Lower Limb Amputation rate of in Diabetic Foot in South Korea: a population based nationwide study
We sincerely thank the editor and reviewers for their insightful and constructive comments. We have considered all comments and responded to them below. We hope that the manuscript is now deemed suitable for publication in The International Journal of Environmental Research and Public Health.
Thank you again for the careful and helpful review.
Reviewer 2’s Comments:
To apply an intervention we need a mechanism (See Mechanism matters, Nature Medicine volume 16, page347, 2010). The authors' suggestions and their review of the literature do little for this (Hong Kong data accepted). Furthermore, the authors, in their reply state: "The studies are expected to have clinical significance as they can be applied directly in daily living, such as management of room air and room temperature in patient’s residence". Such a statement does not mean much. Do we need to heat the patients' environment or cool it? How was the overall relationship between maximum and minimum daily temperatures? I assume they looked at these before looking at DTR. And I assume they did not share this with us as they found no correlation
I could conceptualize low temperatures could cause increased ischaemia, just as the authors conceptualize high temperatures could related to hygiene problems (not that I’m sure why they should increase ischaemia related amputations). But as it is DTR, it is necessary to propose a mechanism why maximum/minimum temperatures to not effect amputation rate but DTR does, e.g. they feel it is hot/cold and behave accordingly even though the temperature has changed to the other extreme. I don’t think an explanation like this is serious.
- Thank you for your helpful comment. I sincerely agree with your question and understand things you are concerning as an orthopedic foot and ankle surgeon. Nevertheless, we still expect this paper will provide significant information to many future readers.
Sudden changes in temperature can induce hemodynamic changes and alter stress on the cardiovascular system in the elderly who have poor ability to maintain in vivo homeostasis [1,2]. In addition, sudden changes in temperature may impair immune function and release of inflammatory mediators and lead to infection. Acute cooling of the body surface due to high diurnal temperature range causes reflex vasoconstriction in the nose and upper airways, and that this vasoconstrictive response can suppress respiratory defenses and cause cold symptoms by converting an asymptomatic subclinical viral infection into a clinically symptomatic respiratory infection. [3]
Although respiratory tract infection is the most common disease in relation to diurnal temperature range, Yin Fei et al. demonstrated the correlation between diurnal temperature range and hand-foot-mouth disease in the paper. Yin Fei et al. correlated diurnal temperature range with the disease by statistical analysis. Although there was no pathophysiological explanation in the paper, the result was sufficiently explained and understood using big data only. [4]
Correlation between DTR and various vascular-related diseases has been analyzed in the manuscript (line 67 and below). Diabetic foot ulcer, as a disease related to peripheral arterial disease and infection, can reasonably be thought to have a correlation with diurnal temperature range.
In fact, currently many nationwide(5,6,7) and world-wide(8,9) studies related to diurnal temperature range and diseases are conducted as big data study. Indeed, most of the papers published nowadays have significant clinical value even in the absence of in vivo studies.
Again, thank you for your supportive review.
- Ge, W.Z.; Feng, X.; Zhao, Z.H.; Zhao, J.Z.; Kan, H.D. Association between diurnal temperature range and respiratory tract infections. Biomedical and Environmental Sciences 2013, 26, 222-225.
- Cao J, Cheng Y, Zhao N, Song W, Jiang C, Chen R, et al. Diurnal temperature range is a risk factor for coronary heart disease death. J Epidemiol 2009;19:328-332.
- Eccles R. Acute cooling of the body surface and the common cold. Rhinology 2002;40:109-114.
- Yin F., Ma Y., Zhao X., Lv Q., Liu Y., Zhang T., Li X. The association between diurnal temperature range and childhood hand, foot, and mouth disease: A distributed lag non-linear analysis. Epidemiology and Infection, 2017, 145(15), 3264-3273.
- Liang, WM., Liu, WP. & Kuo, HW. Diurnal temperature range and emergency room admissions for chronic obstructive pulmonary disease in Taiwan. Int J Biometeorol 53, 17–23 (2009).
- Jingyan Cao, Yuexin Cheng, Ni Zhao, Weimin Song, Cheng Jiang, Renjie Chen, Haidong Kan, Diurnal Temperature Range is a Risk Factor for Coronary Heart Disease Death, Journal of Epidemiology, 2009, 19
- Youn-Hee Lim, Yun-Chul Hong, Ho Kim, Effects of diurnal temperature range on cardiovascular and respiratory hospital admissions in Korea, Science of The Total Environment, Volumes 417–418, 2012, Pages 55-60.
- Kim, Jayeun, et al. "Comprehensive approach to understand the association between diurnal temperature range and mortality in East Asia." Science of the Total Environment 539 (2016): 313-321.
- Lee, WH., Lim, YH., Dang, T.N. et al. An Investigation on Attributes of Ambient Temperature and Diurnal Temperature Range on Mortality in Five East-Asian Countries. Sci Rep 7, 10207 (2017).
Materials and Methods / Discussion
- The main outcome, diabetic foot amputation related with peripheral artery disease defined only with ischemia related interventions. But according to the methods, diabetes was not a criterion. So basically this issue is relating amputations for ischemia and not for diabetes.
- Thank you for the comment. As we mentioned in 1 Study population and 2.3 Definition of diabetic foot amputation/ulcer part in Materials and Methods section, the diabetes foot amputation/ulcer outcome defined by the treatment or surgical code of diabetic patients after diagnosis of diabetes. In addition, the treatment or surgical codes included in the definition of diabetes development include some angioplasty, metallic stent, and atherectomy, but are primarily included in the surgical operation amputation.
(Methods, page 3, lines 118-121.)
~and amputation of extremities (N0571-N0575). In addition, amputation after index date of DM was considered as the occurrence of an event and the earliest amputation was classi-fied as the index event.
2.Looking at the results, the effect sizes for most northern areas were around 2%, with the exception of Incheon and Gangwon-do, about 5%. By the way why not refer to northern & southern areas rather than latitude? And then the southern areas have an opposite or no effect. So would the authors recommend an intervention only in the northern regions?
It is not as if the weather is so different between northern and southern regions in South Korea (~250 km) to explain this difference or permit this type of data manipulation. Furthermore, I might suggest that the northern regions are slightly different socioeconomically, a possible confounder not looked at.
If I were an officer in Seoul with funds for preventing amputations related to PVD or to diabetes, I would probably want to invest in one of the interventions with proven efficacy. A 2% increase in some regions does not provide reason for a nationwide intervention.
All in all, this finding is at most a curiosity, not supported by a mechanism, for a diagnosis not correctly defined and based on doubtful statistics at the best.
- Thank you for the comment. We agree with the reviewer, and acknowledge that the results of the study have not been consistent region by region and that in some regions the opposite results can reduce the robustness and reliability of the results. It’s hard to interpret the adverse effect of DTR, but there are some possible reasons; First, as we mentioned in the limitation section, it may be the result of inconsistency with the daily DTR exposed to individuals because we have seen a link between the daily DTR and the results of the care facility's area. Second, in current study, we could not include seasonal factors as an analytical variable. Since daily DTRs generally vary in distribution according to seasons, increasing the number of daily DTRs by one unit over the seasons may have different meanings. This limitation would have related with estimated effect size as well. Third, . We further discussed in limitation section considering reviewer’s comments as below.
- In addition, the subgroup analyses for region (Region 1 and 2) was conducted to compare based on latitude in South Korea. Although the socioeconomic status of individuals or region has not been included in the analytical variables in this study, we are planned to include in our future NHIS studies.
(Discussion, page 10, lines 237-241.)
First, since HIRA provides only the location of medical institutions where the amputation was performed, DTR also calculated based on the location of the medical institution, not the individual’s resident, the correlation may be low if patients traveled a long distance for surgery. Therefore, the actual DTR acting on the patient may be different from the measurement because the measuring station is administratively located in each region.
(Discussion, page 10, lines 243-245.)
Third, in current study, we could not include seasonal factor in the analysis. Because the distribution of DTR could related with seasonality, we planned to consider season-ality in future investigation.
(Discussion, page 10, lines 253-258.)
In this study, we have not found statistically significant or robust results of DTR on diabetic foot amputation or ulcers. Nevertheless, this study is meaningful in that it is the first study in Korea to explore the link between DTR and diabetes foot amputation. Based on the current result as pilot study, we will design future study could serve as a basis for policy establishment of managing diabetic foot patients by identifying the re-lationship between DTR and the amputation rate in diabetic foot patients.
- The authors chose not to share the raw data with us. That is their right, but not customary nowadays, when a journal supports/recommends.
- Thank you for the comment. We modified our data availability statement in our Patents section as below.
(Acknowledgement, page , lines .)
The data are distributed to registered user through the official website of HIRA Healthcare Bigdata Hub (https://opendata.hira.or.kr/home.do). After the evaluation of research proposal by HIRA review committee, registered user can receive special access privileges to the data.